# Climatic Zone and Soil Properties Determine the Biodiversity of the Soil Bacterial Communities Associated to Native Plants from Desert Areas of North-Central Algeria

**DOI:** 10.3390/microorganisms9071359

**Published:** 2021-06-23

**Authors:** Elisa Bona, Nadia Massa, Omrane Toumatia, Giorgia Novello, Patrizia Cesaro, Valeria Todeschini, Lara Boatti, Flavio Mignone, Houda Titouah, Abdelghani Zitouni, Guido Lingua, Francesco Vuolo, Elisa Gamalero

**Affiliations:** 1Dipartimento di Scienze e Innovazione Tecnologica, Università del Piemonte Orientale, Piazza San Eusebio 5, 13100 Vercelli, Italy; valeria.todeschini@uniupo.it; 2Dipartimento di Scienze e Innovazione Tecnologica, Università del Piemonte Orientale, Viale T. Michel 11, 15121 Alessandria, Italy; giorgia.novello@uniupo.it (G.N.); patrizia.cesaro@uniupo.it (P.C.); laramv.boatti@gmail.com (L.B.); flavio.mignone@uniupo.it (F.M.); guido.lingua@uniupo.it (G.L.); elisa.gamalero@uniupo.it (E.G.); 3Agro-Pastoralism Research Center (APRC), 17018 Djelfa, Algeria; toumatia@yahoo.com; 4Laboratoire de Biologie des Systèmes Microbiens (LBSM), Ecole Normale Supérieure (ENS) de Kouba, 16000 Algiers, Algeria; houda.tit13@gmail.com (H.T.); zitouni_abdelghani@yahoo.fr (A.Z.); 5SmartSeq s.r.l., Spin-Off of the Università del Piemonte Orientale, Viale T. Michel 11, 15121 Alessandria, Italy; 6Sacco s.r.l., Via Alessandro Manzoni 29/A, 22071 Cadorago, Italy; f.vuolo@saccosrl.it

**Keywords:** desert soil, climatic zone, microbiota, holobiont, arid zone, semi-arid zone

## Abstract

Algeria is the largest country in Africa characterized by semi-arid and arid sites, located in the North, and hypersaline zones in the center and South of the country. Several autochthonous plants are well known as medicinal plants, having in common tolerance to aridity, drought and salinity. In their natural environment, they live with a great amount of microbial species that altogether are indicated as plant microbiota, while the plants are now viewed as a “holobiont”. In this work, the microbiota of the soil associated to the roots of fourteen economically relevant autochthonous plants from Algeria have been characterized by an innovative metagenomic approach with a dual purpose: (i) to deepen the knowledge of the arid and semi-arid environment and (ii) to characterize the composition of bacterial communities associated with indigenous plants with a strong economic/commercial interest, in order to make possible the improvement of their cultivation. The results presented in this work highlighted specific signatures which are mainly determined by climatic zone and soil properties more than by the plant species.

## 1. Introduction

According to FAO, soil degradation is defined as “a change in the soil health status resulting in a diminished capacity of the ecosystem to provide goods and services for its beneficiaries” (http://www.fao.org/soils-portal/soil-degradation-restoration/en/, accessed on 13 May 2021). Typically, land degradation is associated to direct or indirect human-induced processes, including anthropogenic climate change such as extreme weather conditions. In particular, drought leads to reduced crop yield, food production, livelihoods, and negatively affects almost 2 billion ha of land worldwide. When land degradation occurs in drylands (including arid, semi-arid, dry sub-humid and hyper-arid areas), it is defined as desertification. Considering that about 3 billion people [1] live in drylands, mainly in South and East Asia, North Africa and the Middle East, and that this value is projected to increase and eventually double by 2050, it is easy to understand why desertification has been a major global issue during the 20th century and will remain one of the most relevant environmental and social problems during the 21st century.

Extending over a surface reaching about 2.38 million km^2^, Algeria is the largest country in Africa where three types of climates coexist: a Mediterranean climate which is typical of the Coastal zones and Northern mountains, a semi-arid climate that occurs in the highlands and an arid climate which is associated with the Sahara desert [2]. Based on the aridity index (AI), defined as the ratio between average annual precipitation amount (P) and potential evapotranspiration amount, Algeria is characterized by semi-arid and arid sites located in the North of the country, and occupying 20% of the territory and hypersaline zones in the center and South of Algeria, representing 80% of the country. As a consequence, only 8.5 million ha of the Algerian land (3.5%) are used for agriculture (http://www.fao.org/3/a-bc290e.pdf, accessed on 13 May 2021). However, amongst the Arab countries, Algeria shows the highest biodiversity in plant species, with 3164 plant species, 178 of which are unique to this territory [3]. Several autochthonous plant species are well known as medicinal plants. They have been used for centuries to treat a wide range of diseases according to their own ethno-medical tradition. As an example, populations living in the Sahara desert (South of Algeria) are still almost completely reliant on traditional healers for health purposes and for the treatment of pain [3,4]. In Algeria, the typical vegetation follows North-South climatic gradient; all the autochthonous plants have in common a certain degree of tolerance to aridity, drought and salinity.

In their natural environment, plants live with an impressive amount of different microbial species including archaea, bacteria, fungi and protists. By using the terminology applied for microorganisms inhabiting the human body, these microorganisms are indicated as plant microbiota. These associated bacteria are selected by the release of plant exudates and proliferate outside and inside the plant tissues, affecting plant growth, health, yield as well as the nutritional value of seeds and fruits [5,6,7,8,9,10,11,12,13,14,15]. Thus, they play a relevant role in the improvement of the host’s life and fitness. When plants are exposed to biotic or abiotic stressful conditions they express their own adaptation mechanisms, but also rely on the plant-associated microorganisms in order to cope with the stress and survive. In this context, the plants are now viewed as a “holobiont”, a superorganism compri-sing the plant, the associated microbiota and the multiple interactions occurring among them [16,17,18].

This work had a dual purpose: first of all, to deepen the knowledge of the arid and semi-arid environment, still little known although it has a strong biological potential; se-condly, to characterize the composition of bacterial communities associated with indigenous plants with a strong economic/commercial interest in order to make possible the improvement of their cultivation.

In this context, different economically relevant autochthonous plants in Algeria (Appendix A) [19,20,21,22,23,24,25,26,27,28,29,30,31,32,33,34,35,36,37,38,39,40,41,42,43], were chosen in their natural area in order to characterize their associated bacterial communities. In addition, the composition of the bacterial communities was also related to climatic variables and to the chemical and physical parameters of the soils.

## 2. Materials and Methods

### 2.1. Soil Sampling

Soil sampling was performed during September 2018, in two different Algerian regions (Ghardaïa and Djelfa), in six different provinces (Figure 1) in correspondence of 14 active plant species (Appendix A). Four species (*Cleome arabica*, *Reseda villosa*, *Zilla spinosa*, *Pulicaria undulata*) were sampled in the arid region of Ghardaïa (Figure 1A) and ten species (*Arthrophytum scoparium*, *Astragalus armatus*, *Retama raetam*, *Stipa tenacissima*, *Artemisia herba-alba*, *Salsola tetragona*, *Atriplex halimus*, *Peganum harmala*, *Suaeda fruticosa*, *Thymelaea microphylla*) were sampled in the semi-arid region of Djelfa (Figure 1B). While Ghardaïa region is located in the South of Algeria, where the summer is hot and dry, with mean summer temperature of 36.8 °C and maximum absolute temperature of 46 °C, the Djelfa region is characterized by very cold weather in winter and hot in summer, with strong temperature excursion between night and day (Appendix A). Before sampling, 5 cm topsoil was removed. The plants were active during sampling. After digging pits around the plants, the rhizosphere soil was differentiated by removing the external soil then soil found adjacent to the roots was collected using sterile gloves. In particular, soils associated to the roots (rhizosphere soils) were collected perpendicularly all along the roots to a depth ranging from 5 to 20 cm and it was the same for all the species. The five pits (around 20 × 20 × 15 cm) were dug for each sampled plant, then the soil was mixed to form five biological replicas and stored at 4 °C until taken to laboratory, where the soils were stored at −80 °C until analyses. The final volume of each soil was around 100 cm^3^.

The soil samples were analyzed for physical–chemical properties (Appendix A) using standard procedures reported in detail in Massa et al., (13). The meteorological data related to the monthly average of maximum and minimum temperatures were identified by the two stations of Djelfa and Ghardaïa of the Algerian National Office of Meteorology. ONM collect the climatic data by using automatic stations near to the sampling sites, the collected data were verified and compared to Alessandria’s climatic data using the Google Earth Engine tool. Data regarding temperature, humidity and rainfall are also reported in the supporting information (Appendix A).

### 2.2. Microbiome Characterization

The genomic analysis of the soil microbiome was performed on soil samples collected from 14 autochthonous plants. The genomic DNA was extracted using the DNeasy^®^ Po-werSoil^®^ Kit (Qiagen, Milan, Italy), starting from 0.25 g of soil following the manufacturer’s instructions and quantified by a fluorometric method according to the Qubit^®^ 4.0 Fluorimeter protocol. The preparation of the bacterial 16S DNA libraries was performed using the Microbiota solution B kit (hypervariable regions V3-V6) provided by Arrow Diagnostics srl. (Genoa, Italy), according to the manufacturer’s instructions. The amplicon pool was processed using the MS-103-1003-MiSeq Reagent Nano Kit v2 (500-cycles) kit, supplied by Illumina Inc., using the Phix as internal standard.

### 2.3. Bioinformatic and Statistical Analyses

Bioinformatic analysis workflow is proposed in this paper by the authors for the first time. Obtained raw sequences were processed with the new software MicrobAT (Microbiota Analysis Tool) v. 1.1.0 provided by UPO-SpinOff (SmartSeq srl, Novara, Italy). The software specifies the Phylum, Class, Order, Family, Genus and Species of the bacteria found in the samples and provides reports of the user-selected comparisons. MicrobAT is based on the RDP database and it does not produce OTUs (operational taxonomic units). In particular, obtained sequences, after being filtered for length and quality (data quality evaluation), were aligned against the RDP database and were assigned to a specific species if they meet the following criteria: query coverage ≥80% and similarity ≥97%. From MicrobAT three files can be generated, which were used for statistical analyzes regarding variations within the bacterial communities using the Microbiome-Analyst software (Comprehensive Statistical, Visual, and Meta-Analysis of Microbiome data; https://www.microbiomeanalyst.ca, accessed on 13 May 2021). Before data analysis, a data integrity check was performed.

First data filtering was used in order to identify and remove features that are unlikely to be useful when modeling the data. Features having low count and variance can be removed during the filtration step while those having very few counts are filtered based on their abundance levels (minimum counts 10) across samples (prevalence). Data rarefaction and scaling based methods deal with uneven sequencing depths by bringing samples to the same scale for comparison.

Alpha diversity was characterized by the total number of observed species (richness) and by Shannon and Simpson indexes that, along with the number of species (richness), consider also the abundance of organisms (evenness) to describe the actual diversity of a community. Alpha diversity analysis was performed using the phyloseq package [44]. The results were plotted across samples and reviewed as box plots for each considered group (plant species, sampling site and climatic zone).

Beta diversity analysis was used to compare the diversity of composition between the sampled bacterial communities. This method compares the changes in the pre-sence/absence or abundance of the present species and summarizes these into how ‘similar’ or ‘dissimilar’ the samples are. Each sample gets compared to every other sample generating a distance matrix. The distance between samples was measured using Bray-Curtis distance and Principal Coordinate Analysis (PCoA) was used to visualize these matrices in 2 plot where each point represents the entire microbiome of a single sample. Each axis reflects the percentage of the variation between the samples with the *X*-axis representing the highest dimension of variation and the *Y*-axis representing the second highest dimension of variation. Further, each point or sample displayed on PCoA plots is colored based on either sample group (plant species, sampling site and climatic zone). Moreover, the statistical significance of the clustering pattern in ordination plots can be evaluated using Permutational ANOVA (PERMANOVA). Beta diversity analysis was performed using the phyloseq package [44].

Hierarchical cluster analysis was also performed at phylum level. Two parameters were considered. The first one was the distance measured between samples (Bray-Curtis distance). The other parameter was clustering algorithms, including average linkage result shown as heatmap (distance measure using euclidean and clustering algorithm using ward.D at Phylum level).

Heat tree method was used to compare abundance at phylum level for each pair of sampling sites. Heat tree uses hierarchical structure of taxonomic classifications to quantitatively (median abundance) and statistically (non-parameter Wilcoxon Rank Sum test) depict taxon differences among communities. It generates a differential heat tree to show which phyla are more abundant in the different considered sampling sites. Heat tree ana-lysis is performed using *metacoder* R package, according to Foster [45].

The core microbiome analysis was performed in order to identify core species that remain unchanged in their composition across the whole bacterial community in the different plant species, sampling sites and the two climatic zones. Two parameters were considered. The first one is sample prevalence, which is defined as minimum fractions (percentage) of samples that a species must be observed. The other parameter is the relative abundance (fractions) of a species in order to consider them as a part of the core member. Core microbiome analysis is adopted from the core function in *microbiome* R package. The result of this analysis is represented in the form of heatmap of core taxa where *Y*-axis represent the prevalence level of core features across the detection threshold (Relative abundance) range on *X*-axis.

Moreover, PCA analysis was performed using all the considered parameters, based on the factors plant, sampling site and climatic zone. This analysis was performed using R (v. 3.5.1) [46], in particular, *FactoMineR* [47] and *Factorextra* [48] packages.

Finally, in order to identify the signature associated with the different parameters, Linear Discriminant Analysis Effect Size (LDA-LEfSe) method was applied at species level. This method is specifically designed for biomarker discovery and explanation in high-dimensional metagenomic data [49]. It incorporates statistical significance with biological consistency (effect size) estimation. It performs non-parametric factorial Kruskal-Wallis (KW) sum-rank test to identify species with significant differential abundance with regard to factor of interest (plant species, sampling sites and climatic zone), followed by Linear Discriminant Analysis (LDA) to calculate the effect size of each differentially abundant features. The result consists of all the species with the highest mean and the logarithmic LDA score (Effect Size). Features are considered to be significant based on their adjusted *p*-value. The default *p*-value cutoff was 0.05.

## 3. Results

### 3.1. Soil Properties

The analyzed sampling sites correspond to different combinations of chemical-phy-sical soil characteristics as reported in Appendix A. Soil texture resulted sandy-silty in Metlili, Beni Isguen (Ghardaïa) and in Zaafrane (Djelfa), while the soils of Messaad, Ain Naga and Moudjbara (Djelfa) were silty-sandy. pH ranged from 7.2 to 8.3. In particular, Metlili and Moudjbara were slightly alkaline soils (7.2) while the other samples were sub-alkaline (Ain Naga and Zaafrane—7.8 and 7.9 respectively) and alkaline soils (Beni Isguen and Messaad—8.3 and 8.2 respectively).

The soils of all sampled sites were found to be very poor in organic matter (<0.8%). Moreover, soils of Metlili, Beni Isguen and Zaafrane were moderately calcareous (5–10%), while those of Messaad, Ain Naga and Moudjbara were very calcareous (10–25%). Active limestone was good (2–5%) in Metlili, Messaad and Zaafrane while it was rich (5–10%) in Beni Isguen, Ain Naga and Moudjbara soils. Considering the percentage of nitrogen, the soil of Messaad can be considered poor (0.05–0.07%), those of Ain Naga, Zaafrane and Moudjbara can be considered with an average concentration (0.08–012%) while Metlili and Beni Isguen have a good concentration (0.13–0.24%) of nitrogen. Finally, considering the concentration of available phosphorus, the Zaafrane soil was very scarce (<7%), Me-tlili and Messaad soils had a low (7–14%) content of assimilable phosphorus while soils of Beni Isguen, Moudjbara and Ain Naga were found to have an average (15–20%), good (21–30%) and rich (31–45%) content, respectively.

### 3.2. Biodiversity

A total of 5,036,610 reads were obtained with a mean value of 74,068 reads per sample. After the demultiplexing step, a total of 4,313,231 reads (with a mean value of 63,430 reads per sample) were used for further analysis. The genomic sequences were included in the BioProject PRJNA719273 available in NCBI database https://submit.ncbi.nlm.nih.gov/subs/sra/SUB9344565/overview, accessed on 13 May 2021. The BioProject contains 70 BioSamples with IDs from SAMN18593889 to SAMN18593958.

### 3.3. Community Profiling

The evaluation of the three alfa diversity estimators was performed at the species level according to the plant (Figure 2A), the sampling sites (Figure 2B) and climatic zones (Figure 2C).

The number of observed species in the rhizosphere microbiota differed significantly according to the plant species (*p* = 0.01107) and the sampling site (*p* = 0.00001). The highest number of observed species was recorded in the rhizosphere of *S. tenacissima* (108), *A. herba alba* (108) and *C. arabica* (107), while the lowest amount of bacterial species occurred in the rhizosphere of *A. armatus* (101), *A. scoparium* (102) and *T. microphylla* (100) (Figure 2A left). By considering the sampling site, the microbiota associated to plants native of Ain Naga and Moudjbara accounts for the highest number of bacterial species (108), while that associated to plants growing in Messaad and Zaafrane showed the lowest number of observed species (103 and 104, respectively) (Figure 2B left). The amount of the observed species found in the plant rhizosphere did not change significantly (*p* = 0.195) according to the climatic zone (Figure 2C left).

Shannon’s diversity index differed significantly according to the plant species (*p* = 0.0011), the sampling site (*p* = 0.00011) and the climatic zone (0.0061). The rhizosphere of *S. tenacissima* (3.23), *P. undulata* (3.13) and *C. arabica* (3.10) hosted microbiota with the hi-ghest biodiversity. On the contrary, the rhizosphere microbiota of *A. scoparium* and *A. armatus* showed the lowest values of the Shannon’s index (2.24 and 2.30, respectively) (Figure 2A center). The microbiota associated to plants from Ain Naga and Beni Isguen showed the highest biodiversity indexes (3.23 and 3.03, respectively), while that asso-ciated to plants native of Messaad and Zaafrane were characterized by the lowest value of Shannon’s index (2.37 and 2.70, respectively) (Figure 2B center). Finally, the biodiversity of the rhizosphere microbiota of plant species typically associated to arid zones (2.96) was higher than that observed in the rhizosphere of plants growing in semi-arid zones (2.67) (Figure 2C center).

The Simpson’s biodiversity index changed significantly according to the plant species (*p* = 0.0011), the sampling site (*p* = 0.00012) and the climatic zone (0.0046). The rhizosphere microbiota of *P. undulata* (0.885), *S. tenacissima* (0.871) and *C. arabica* (0.843) showed the highest biodiversity, while the rhizosphere microbiota of *S. tetragona* (0.705), *A. scoparium* (0.694) and *A. armatus* (0.680) was characterized by the lowest values of the Simpson’s index (Figure 2A right). The microbiota associated to plants sampled in Ain Naga (0.87) and Beni Isguen (0.86) showed the highest biodiversity index, while the rhizosphere bacterial community of plants from Messaad and Zaafrane were characterized by the lowest value of Simpson’s index (0.71 and 0.76, respectively) (Figure 2B right). The biodiversity of the rhizosphere microbiota of plants growing in arid zones (0.83) was higher than that measured in the rhizosphere of plants native to semi-arid zones (0.76) (Figure 2C right).

Beta diversity (the comparison of bacterial communities based on their composition) provides a measure of the distance or dissimilarity between each sample pair. Principal Coordinates Analysis (PCoA), performed on the recorded species (Figure 3), shows that the first axis explains 43.1% of the differences and the second one 25.1%. The overall composition of the soil microbiota, considered at species level, was significantly affected by plant species, sampling site and climatic zone, as determined by non-parametric multivariate analysis of variance testing (PERMANOVA; *p* < 0.001 for all factors). This grou-ping was also confirmed by the analysis of similarity (ANOSIM) test, which evaluates significance of sample grouping (Plant-R = 0.694, Sampling site-R = 0.461, Climatic zone-R = 0.300; *p* < 0.001 for all factors).

Since our data indicates that the sampling sites (different soil properties) and climatic zones are the main factors driving the biodiversity of the rhizosphere microbiota of the 14 plant species considered, we decided to construct a heatmap representing the gradient common core according to the classification of the sampling sites (Appendix A) and the climatic zones (Appendix A) at phylum and species levels, respectively. Unclassified Actinobacteria resulted as the dominant species in all the sampling sites with the exception of Ain Naga and Moudjbara (Appendix A).

A total of 10 and 11 phyla (Appendix A) (corresponding to 23 and 21 bacterial species—Appendix A) were identified respectively as components of the core microbiota of the rhizosphere of plants growing in arid (Ghardaïa) and semi-arid (Djelfa) climatic zones.

Seventeen bacterial species recognized as component of the core microbiota were shared between arid and semi-arid zones. Bacterial species belonging to unclassified Bacteria and unclassified Actinobacteria were the most spread in the rhizosphere of the plants growing in both arid and semi-arid zones. In the rhizosphere of the four plants growing in arid climatic zones the top 10 species were completed by species belonging to unclassified Actinomycetales, unclassified Planctomycetaceae, unclassified Chitinophagaceae (Bacteroidetes), unclassified Gaiella (Actinobacteria), unclassified Rhizobiales (Proteobacteria), unclassified Chloroflexi, unclassified WPS1 genera incertae sedis and unclassified Anaerolinaceae. A different situation was observed in the microbiota of the rhizosphere of the ten plants growing in semi-arid zones where the top 10 bacterial species, after unclassified Bacteria and unclassified Actinobacteria, were unclassified WPS1 genera *incertae sedis*, unclassified Actinomycetales, unclassified Chitinophagaceae (Bacteroidetes), unclassified Planctomycetaceae, unclassified Rubrobacter (Actinobacteria), unclassified Rhizobiales, unclassified alpha-Proteobacteria and unclassified Gemmatimonas.

### 3.4. Microbiota Comparison

The heatmap representing the abundance of the different bacterial phyla (Figure 4 and Appendix A) showed a clear distribution according to the climatic zones and to the sampling sites.

While Armatimonadetes, Actinobacteria, Firmicutes, Acidobacteria, Chloroflexi, Planctomycetes, members of the candidate division WPS2 and Nitrospirae were typically associated to arid zones, members of candidate division of WPS1, Gemmatimonadetes, Verrucomicrobia, Parcubacteria, Proteobacteria, Bacteroidetes, candidatus Saccharobacterium were associated to semi-arid zones. Moreover, unclassified Archaea were prevalent in arid zones (Appendix A). Figure 5A represents an overview of the five principal components determining the variability of the plant microbiota. Principal component analysis (PCA) of the phyla occurring in the plant rhizosphere were performed according to plant species (Figure 5B), sampling site (Figure 5C) and climatic zones (Figure 5D). Altogether, axis 1 and 2 describe 45% of the sample variability. This analysis highlights that the bacterial phyla clustered more according to the climatic zones than to plant species and confirm the phyla distribution already observed in Figure 4.

### 3.5. Signature

Linear Discriminant Analysis Effect Size (LDA-LEfSe) was applied at species level in order to determine the signature: this information allowed us to identify the bacterial species with significant differential abundance according to climatic zone (Table 1) and sampling sites (Appendix A). The climatic zone parameter determined the most marked distribution of abundances of the bacterial species: at the top of the Table 1 are reported the bacterial species more relevant in the semi-arid zone while at the bottom those important for arid one. Table 1 highlighted that Actinobacteria, Bacteroidetes and Proteobacteria were more represented in semi-arid zone in respect to arid one, where Actinobacteria (different genera), Proteobacteria (different genera), Chloroflexi, Planctomycetes and Firmicutes were the most prevalent. Specific bacterial signature was outlined in the different climatic zones. In fact, semi-arid zone was characterized by Microbacteriaceae, *Rubrobacter* sp., *Ilumatobacter* sp. (from Actinobacteria), Cytophagales, Flavobacteriaceae and *Ohtaekwangia* sp., (from Bacteroidetes), Alphaproteobacteria, Gammaproteobacteria, Sphingomonadales Phyllobacteriaceae, Caulobacteraceae, Xanthomonadaceae, Erythrobacteraceae, *Phaselicystis* sp., *Steroidobacter* sp. (from Proteobacteria) while arid zone was characterized by: Actinomycetales, Acidimicrobiales, Pseudonocardiaceae, Micromonosporaceae, *Aci-diterrimonas* sp., *Solirubrobacter* sp., *Thermoleophilum* sp., *Euzebya* sp. (from Actinobacteria), Anaerolineaceae, *Sphaerobacter* sp., *Litorilinea* sp. (from Chloroflexi), Planctomycetaceae, *Pirellula* sp. and *Gemmata* sp. (from Planctomycetes) and Betaproteobacteria, Deltaproteobacteria, Burkholderiales, Rhodospirillales, Cystobacteraceae, *Rubellimicrobium* sp., *Microvirga* sp., *Nitrospira* sp. and Sphingomonas (from Proteobacteria). It is much more complex to delineate a specific signature associated with specific soil parameters.

All possible signatures are shown in Appendix A. The clearest and most worthy associations are the decrease in Acidobacteria associated with the pH variation in the two sites of the arid zone (Beni Isguen vs. Metlili, 8.3 vs. 7.2) accompanied by the increase in Bacteroidetes (Cytophagaceae and Aderibacter) and in Actinobacteria (Aciditerrimonas, Euzebya, Gaiella, Nocardioides, Thermoleophilum). Furthermore, the association of Chloroflexi with a higher concentration of available phosphorus is important, which determines a more marked presence of Anaerolineaceae, Litorilinea and Sphaerobacter.

## 4. Discussion

The present work describes the bacterial communities associated to the soils of different desert regions of North-Central Algeria. These soils are characterized by a very scarce vegetation cover mainly represented by native species adapted to these extreme conditions. The results highlighted that climatic zone (arid and semi-arid) and soil pro-perties determined the biodiversity of the microbiota associated to fourteen common native plants from desert areas of North-Central Algeria. The sampling sites outlined not only a geographical position but above all the set of chemical-physical soil properties. Therefore, the sampling site can be considered as a parameter related mainly to the substrate. The analyzed soils appeared to be very poor in organic matter, which is not surprising considering the desert condition associated both to arid and semi-arid areas. Total organic carbon, electric conductivity, pH and total phosphorus were the dominant factors to affect the microbial communities associated to desert environment [50]. In fact, the highest number of observed species (alpha-diversity) was found in the two P richest soils, Ain Naga and Moudjbara, while Messaad and Zaafrane that have lower P concentrations, showed the lowest number of observed species. Considering also Shannon and Simpson indexes, the highest values were recorded in Ain Naga (associated to high P concentration) and the lowest were Messaad and Zaafrane. Moreover, the alpha-diversity in terms of biodiversity indices (Shannon and Simpson index) varied in a statistically significant way for the other two considered factors as well as beta-diversity. The statistically signi-ficant difference observed for alpha and beta diversity is determined by the different pre-sence of the species belonging to the signature that will be discussed later.

In addition, considering another important soil parameter, such as active limestone, whose concentration was the greatest in the highest P concentration sites (Ain Naga, Moudjbara and Beni Isguen), it could not have a positive effect on the availability of this fertilizing element but on the contrary it could be an indication of a negative situation due to a general insolubilization of nutrients [51]. The soils considered in this work can be divided into three categories: slightly alkaline, subalkaline and alkaline. These character-ristics could affect the composition of the bacterial communities particularly concerning, as suggested before, the Actinobacteria and also the Acidobacteria. In the arid zone, Met-lili and Beni Isguen had two different soil alkalinities that can determine the difference in Actinobacteria. In fact, in Beni Isguen soil (pH 8.3) the concentration of Actinobacteria (*Aciditerrimonas* sp., *Euzebya* sp., *Gaiella* sp., Nocardiodes and *Thermoleophilum* sp.) was higher than in Metlili (pH 7.2), while the concentration of Acidobacteria (*Balstocatella* sp.) followed the opposite trend. In slightly alkaline soils (pH 7 to 7.5), Actinomycetes mainly develop and they are able to compensate the low activity of fungi and bacteria in periods of water scarcity, typical of this environment [51]. The same soil conditions contributed to the higher abundance of Bacteroidetes (sevenfold higher in Beni Isguen than in Metlili) (*A-dhaeribacter* sp., Cytophagaceae), confirming the trend observed by Khan and coworkers [52] in the rhizosphere of different medicinal plants in arid land. In the literature other authors reported that Bacteroidetes are able to produce ACC deaminase in rhizosphere [53,54]. On the other hand, indole acetic acid (IAA) was reported to be higher in the rhizosphere of *Adenium obesum*. This could also be attributed to the abundance of Actinobacteria, that are known to produce IAA, as previously shown by literature papers [55,56]. Studies on various desert environments, reveal also both the antimicrobial [2,57] and the plant growth promoting properties [58] for the communities of Actinobacteria. These bacterial performances could explain the survivability potentials of the plant species present in the analyzed sites, during low water and nutrient availability.

Moreover, Planctomycetes (Planctomycetaceae, *Pirellula* sp. and *Gemmata* sp.) were part of the signature of the arid zone. This information is partially in contrast with what has been reported by Vásquez-Dean and coworkers [59], but it must be taken into account that the two works considered different conditions and types of soil. As presented in Fierer and coworkers [60], Actinobacteria and Bacteroidetes are generally more abundant in desert soils than in forests, grasslands, and tundra, while Verrucomicrobia and Acidobacteria show the opposite trend. Chloroflexi, Firmicutes, and Gemmatimonadetes were also found in the Algerian soils, confirming the literature data [60,61], with relative abundances representing less than 5% of the identified sequences.

Investigations regarding bacterial diversity in the semi-arid zone have revealed, as described in the signature paragraph, the predominance of Actinobacteria, Bacteroidetes and Proteobacteria partially confirming the results obtained in the rhizosphere of cactus plants in central Mexico [62]. Fierer and coworkers [60] showed that all the bacterial communities described in different biomes (cold deserts, warm deserts, forests, grasslands and tundra) were dominated by Acidobacteria, Actinobacteria, Bacteroidetes, Proteobacteria, Verrucomicrobia, and Gemmatimonadetes, phyla known to be relatively abundant and omnipresent in the soil. The desert soils are very dry, scanty in nutrients, generally with a basic pH, higher than other biomes. Moreover, the paucity (or complete absence) of plant biomass reduces the inputs of organic carbon useful for bacterial metabolism. In contrast, when the amount of organic matter is greater in soil, the stability and diversity of the microbiota is higher too [63]. For example, Gao and coworkers [64], in a study concerning the effect of aridity and dune type (in a desert of Northern China), showed that Firmicutes, Actinobacteria, Proteobacteria, and Acidobacteria were the dominant phyla in all samples of the rhizosphere of *Caragana microphylla*. The increased abundance of Actinobacteria in the rhizosphere soil was, as suggested before, mainly caused by the decreased soil pH due to rhizodeposition [64]. The authors stated that the structure of the rhizosphere bacterial community was modulated mainly by soil total organic C, total N, Na^+^, and total P while total organic carbon, electronic conductivity, pH, and total pho-sphorus were the dominant factors to affect the bacterial communities associated to the different dunes. Pereira et al. [65] confirms that the low density of bacteria in the desert, compared to a soil not subjected to water stress, can be due to the previous reported phy-sical-chemical characteristics of the arid soil.

## 5. Conclusions

As well documented in the literature, bacterial populations associated with plant roots are strongly influenced by the production of root exudates. On the other hand, plant genetics also have a role in the selection of the associated communities. However, our results demonstrated that in highly stressful environmental conditions, such as in desert environments, the extreme climatic conditions and the composition of the substrate are the main variables affecting the selection and recruitment of bacterial populations. In fact, specific signatures associated with the different conditions were identified for the first time, filling a gap in the current literature. Moreover, it’s our opinion that the plant holobiont takes origins from an adaptation process to these extreme conditions where the bacterial component of this association plays a decisive role in favoring the survival of the plant.

## Figures and Tables

**Figure 1 microorganisms-09-01359-f001:**
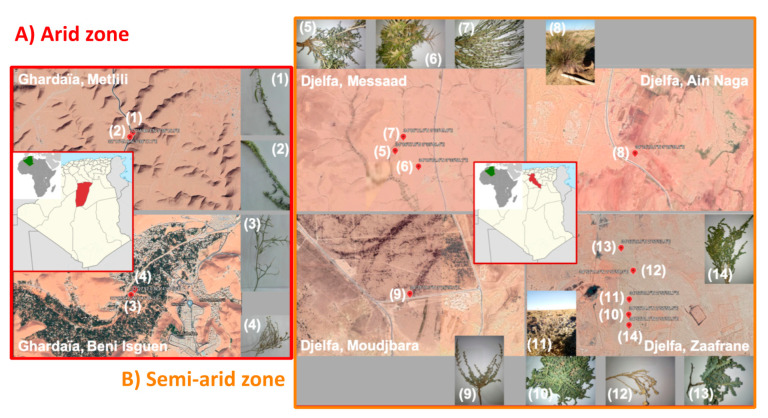
Soil sampling sites. (**A**) Two soil sampling sites in the arid region of Ghardaïa (Algeria): Metlili and Beni Isguen. The sampled soils were associated to the roots of different plants: (1) *Cleome arabica*, (2) *Reseda villosa*, (3) *Zilla spinosa*, (4) *Pulicaria undulata*. (**B**) Four soil sampling sites in the semi-arid region of Djelfa (Algeria): Messaad, Ain Naga, Moudjbara and Zaafrane. The sampled soils were associated to the roots of different plants: (5) *Arthrophytum scoparium*, (6) *Astragalus armatus*, (7) *Retama raetam*, (8) *Stipa tenacissima*, (9) *Artemisia herba-alba*, (10) *Salsola tetragona*, (11) *Atriplex halimus*, (12) *Peganum harmala*, (13) *Suaeda fruticose*, (14) *Thymelaea microphylla*. The numeric labels and the pictures of the different plants are shown in the figure.

**Figure 2 microorganisms-09-01359-f002:**
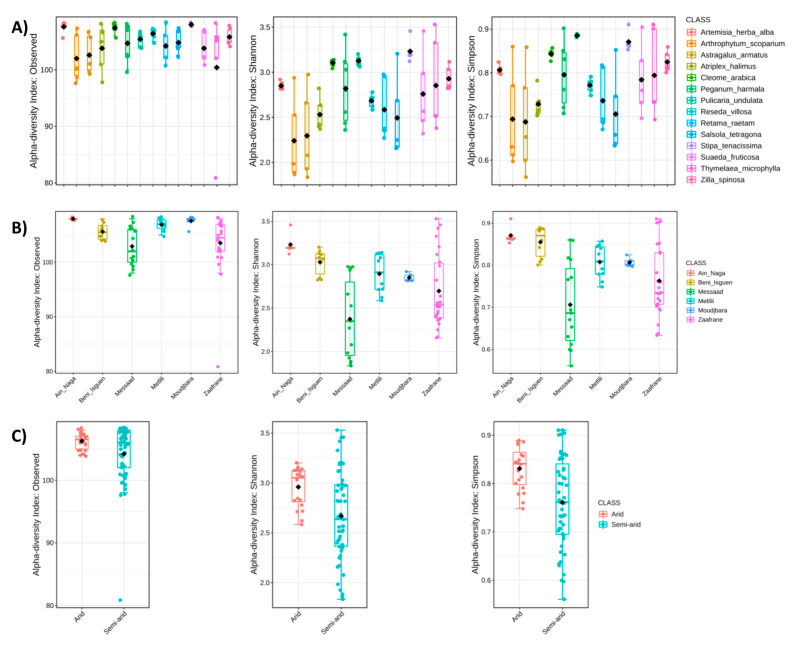
Alfa diversity analysis at species level. (**A**) Number of observed species (*p*-value 0.0111), Shannon’s Index (*p*-value 0.0011) and Simpson’s index (*p*-value 0.0011) in the different plant species. (**B**) Number of observed species (*p*-value 0.0001), Shannon’s Index (*p*-value 0.0001) and Simpson’s index (*p*-value 0.0001) in the different sampling sites. (**C**) Number of observed species (*p*-value 0.1949), Shannon’s Index (*p*-value 0.0061) and Simpson’s index (*p*-value 0.0046) in the two climatic zones. *p*-value cut-off for significance is 0.05. In the figure, black dot indicated mean value while the colored line represented the median value. Alpha diversity analysis was performed using the phyloseq package of MicrobiomeAnalyst.

**Figure 3 microorganisms-09-01359-f003:**
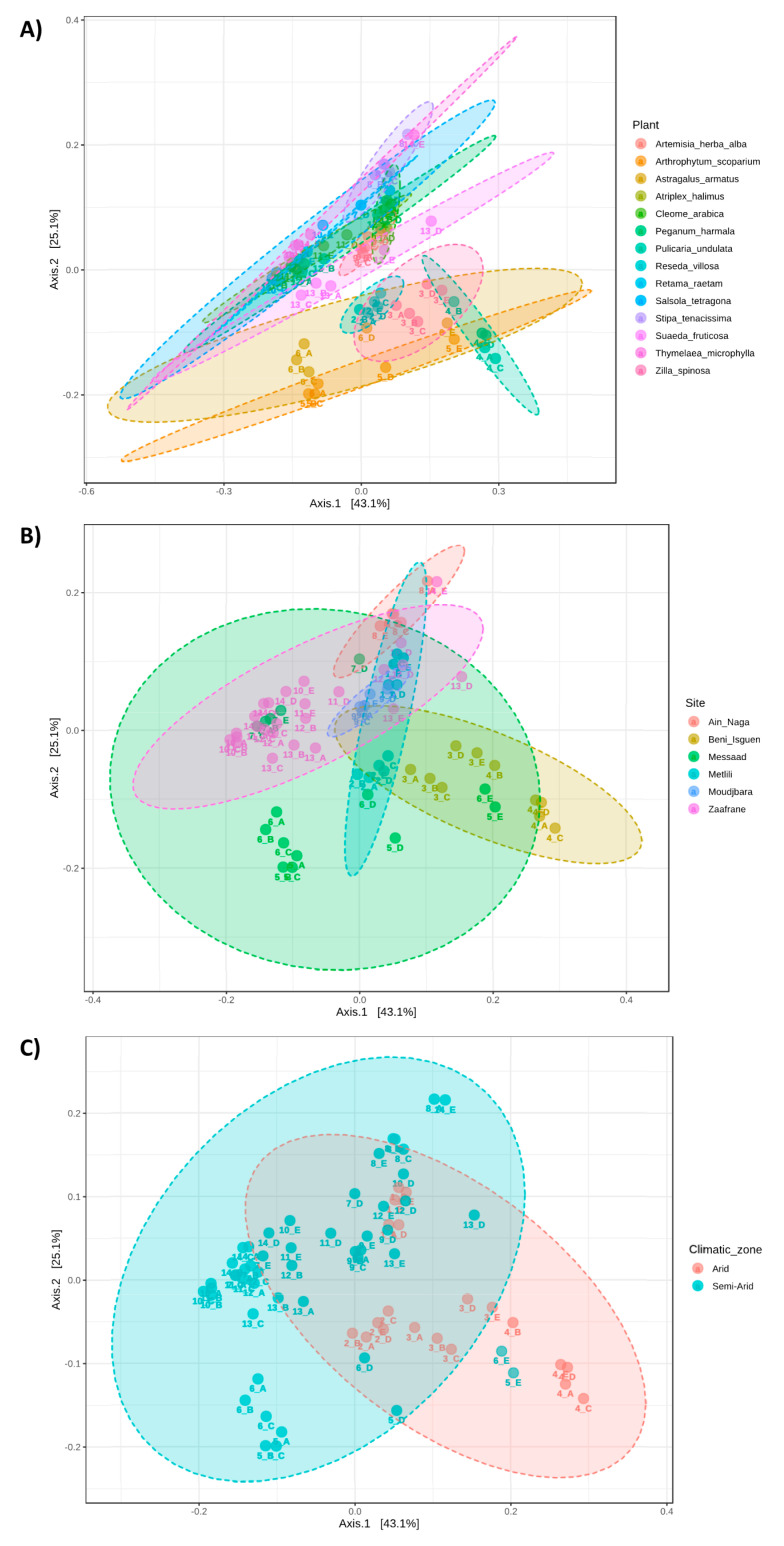
Beta diversity analysis at species level. Principal Coordinate Analysis (PCoA) based on Bray–Curtis metrics shows the dissimilarity of bacterial communities in the different soils according to (**A**) plant species (*p*-value < 0.001), (**B**) sampling site (*p*-value < 0.001), (**C**) and climatic zone (*p*-value < 0.001). Beta diversity analysis was performed using MicrobiomeAnalyst.

**Figure 4 microorganisms-09-01359-f004:**
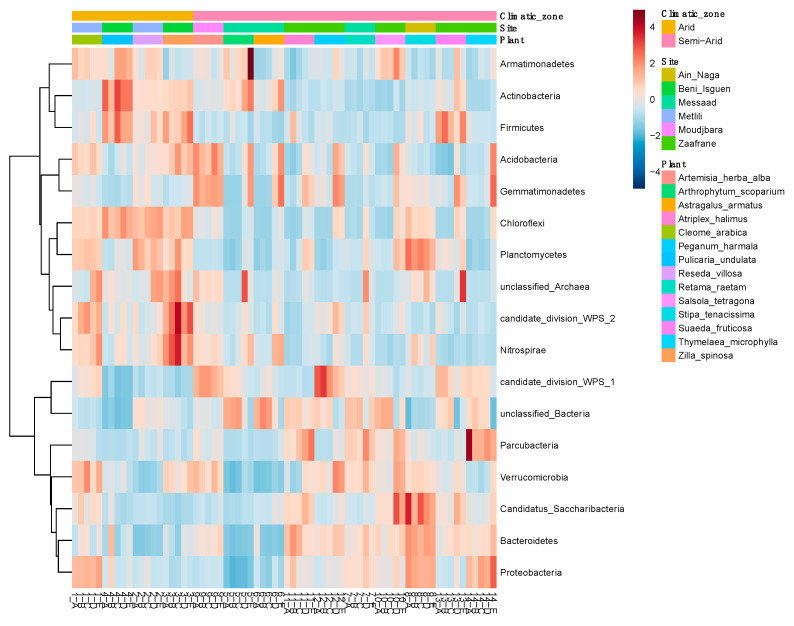
Heatmap at phylum level. Clustering result shown as heatmap (distance measure using euclidean and clustering algorithm using ward.D) at phylum level. Hierarchical clustering is performed with the hclust function in *stat* package of MicrobiomeAnalyst.

**Figure 5 microorganisms-09-01359-f005:**
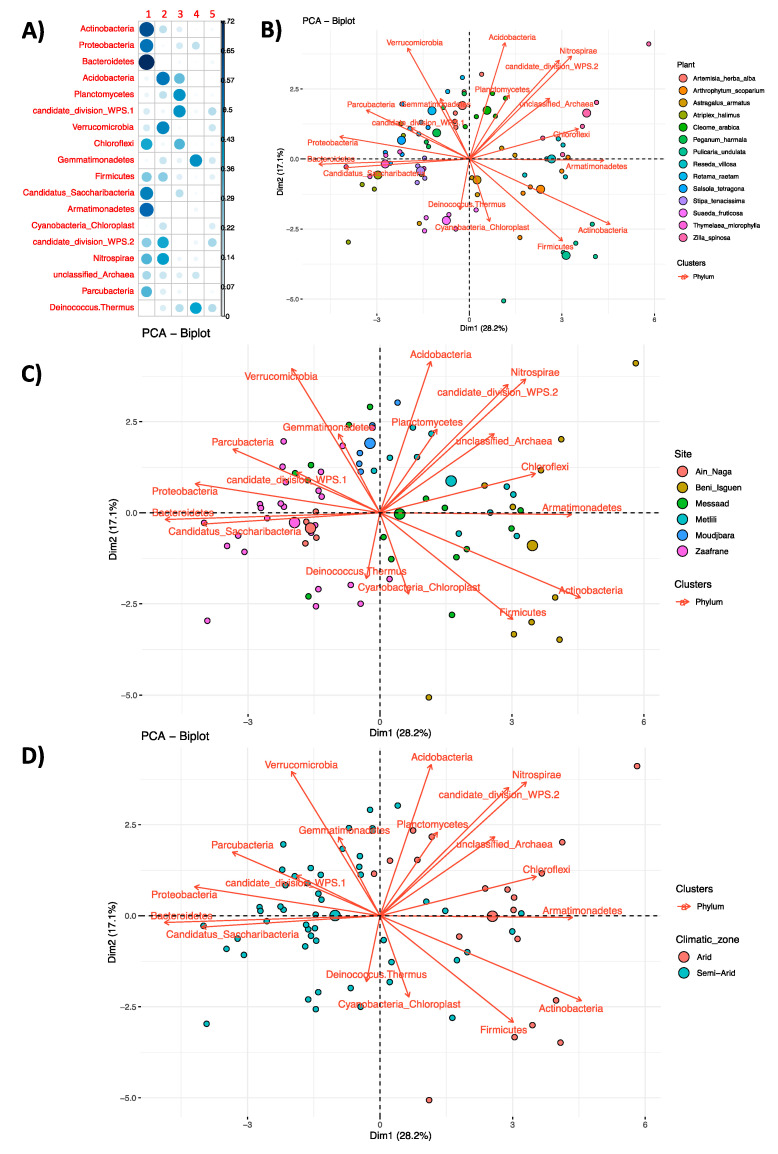
PCA. (**A**) Cos2 contribution of different phyla on PCA dimensions (1, 2, 3, 4, 5). PCA graphs according to plant, sampling site and climatic zone factors. (**B**) PCA-Biplot at phylum level according to plant species; (**C**) PCA-Biplot at phylum level according to sampling site; (**D**) PCA-Biplot at phylum level according to climatic zone. Each bigger dot represents the mean value for each considered parameter while each little dot represents each considered sample.

**Table 1 microorganisms-09-01359-t001:** Signature. LEfSe (Linear discriminant analysis Effect Size) at species level according to climatic zone. Light gray indicates species less present in the arid zone than in the semi-arid one and red indicates a presence less than 50%. Dark gray indicates species more present in arid zone than in the semi-arid one and green indicates a presence more than 200%. LEfSe results using non-parametric factorial Kruskal-Wallis (KW) sum-rank test. Adjusted *p*-value cutoff = 0.05 and LDA score = 1.0. LEfSe analysis was performed with MicrobiomeAnalyst.

Species	Phylum	*p*-Values	LDA Score	Arid/Semi-Arid %
Unclassified Acidobacteria Gp4	Acidobacteria	0.0270070	3.95	64%
unclassified Rubrobacter	Actinobacteria	0.0003125	4.65	56%
unclassified Microbacteriaceae	Actinobacteria	0.0151520	3.63	45%
unclassified Ilumatobacter	Actinobacteria	0.0162540	3.67	42%
unclassified Armatimonas1Armatimonadetes gp1	Armatimonadetes	0.0019786	3.64	54%
unclassified Ohtaekwangia	Bacteroidetes	0.0026650	4.64	48%
unclassified Cytophagales	Bacteroidetes	0.0270060	3.51	40%
unclassified Bacteroidetes	Bacteroidetes	0.0000008	4.81	24%
unclassified Flavobacteriaceae	Bacteroidetes	0.0000008	4.25	2%
unclassified Saccharibacteria genera incertae sedis	Candidatus Saccharibacteria	0.0006425	4.34	49%
unclassified Parcubacteria genera incertae sedis	Parcubacteria	0.0014554	3.44	31%
unclassified Steroidobacter	Proteobacteria	0.0055396	3.45	70%
unclassified Alphaproteobacteria	Proteobacteria	0.0007742	4.51	65%
unclassified Gammaproteobacteria	Proteobacteria	0.0053258	4.14	62%
unclassified Sphingomonadaceae	Proteobacteria	0.0019786	4.43	61%
gamma proteobacterium SA29 B	Proteobacteria	0.0183520	3.38	48%
unclassified Phaselicystis	Proteobacteria	0.0005251	3.27	42%
unclassified Sphingomonadales	Proteobacteria	0.0000782	3.94	41%
unclassified Phyllobacteriaceae	Proteobacteria	0.0040260	3.30	39%
unclassified Caulobacteraceae	Proteobacteria	0.0000018	3.72	25%
unclassified Xanthomonadaceae	Proteobacteria	0.0003989	4.15	24%
unclassified Erythrobacteraceae	Proteobacteria	0.0000002	4.07	11%
unclassified Bacteria	unclassified Bacteria	0.0038659	5.60	83%
unclassified WPS 1 genera incertae sedis	unclassified Bacteria	0.0000185	4.89	52%
unclassified Spartobacteria genera incertae sedis	Verrucomicrobia	0.0006130	3.83	52%
unclassified Verrucomicrobia	Verrucomicrobia	0.0001712	4.22	52%
unclassified Acidobacteria Gp3	Acidobacteria	0.0000050	−4.09	282%
unclassified Acidobacteria	Acidobacteria	0.0075492	−3.78	134%
unclassified Gaiella	Actinobacteria	0.0000037	−4.99	446%
unclassified Aciditerrimonas	Actinobacteria	0.0000029	−4.31	369%
unclassified Solirubrobacter	Actinobacteria	0.0000050	−3.89	284%
unclassified Thermoleophilum	Actinobacteria	0.0000103	−3.69	278%
unclassified Actinobacteria	Actinobacteria	0.0000047	−5.42	200%
unclassified Euzebya	Actinobacteria	0.0244220	−3.50	180%
unclassified Solirubrobacterales	Actinobacteria	0.0000536	−4.31	159%
unclassified Pseudonocardiaceae	Actinobacteria	0.0156950	−3.64	142%
unclassified Actinomycetales	Actinobacteria	0.0015237	−4.63	131%
unclassified Micromonosporaceae	Actinobacteria	0.0081443	−3.89	130%
unclassified Acidimicrobiales	Actinobacteria	0.0081443	−4.00	127%
unclassified Archaea	Archea	0.0019786	−3.57	184%
unclassified Armatimonadetes gp4	Armatimonadetes	0.0000000	−3.86	314%
unclassified Adhaeribacter	Bacteroidetes	0.0189940	−3.69	242%
*Adhaeribacter aquaticus (T)*	Bacteroidetes	0.0213760	−3.45	203%
unclassified Anaerolineaceae	Chloroflexi	0.0000000	−4.84	630%
unclassified Chloroflexi	Chloroflexi	0.0000220	−4.59	194%
unclassified Sphaerobacter	Chloroflexi	0.0003611	−3.52	175%
unclassified Litorilinea	Chloroflexi	0.0022080	−3.07	147%
unclassified Bacillales	Firmicutes	0.0000010	−3.92	215%
unclassified Pirellula	Planctomycetes	0.0000001	−4.26	309%
unclassified Planctomycetes	Planctomycetes	0.0000405	−3.40	205%
unclassified Gemmata	Planctomycetes	0.0031487	−3.29	166%
unclassified Planctomycetaceae	Planctomycetes	0.0069934	−4.56	129%
unclassified Rubellimicrobium	Proteobacteria	0.0000050	−4.34	583%
unclassified Burkholderiales	Proteobacteria	0.0000042	−4.27	273%
unclassified Cystobacteraceae	Proteobacteria	0.0018147	−3.82	253%
unclassified Microvirga	Proteobacteria	0.0006848	−4.09	225%
unclassified Nitrospira	Proteobacteria	0.0000246	−3.58	223%
unclassified Deltaproteobacteria	Proteobacteria	0.0000233	−3.87	181%
unclassified Rhodospirillales	Proteobacteria	0.0004393	−3.49	153%
unclassified Sphingomonas	Proteobacteria	0.0147010	−3.66	140%
unclassified Betaproteobacteria	Proteobacteria	0.0307510	−4.16	125%
unclassified WPS 2 genera incertae sedis	unclassified Bacteria	0.0000246	−4.09	216%
unclassified Gp3	unclassified Bacteria	0.0000308	−3.79	199%
unclassified Subdivision3 genera incertae sedis	unclassified Bacteria	0.0001994	−4.49	175%
unclassified Gp7	unclassified Bacteria	0.0008490	−4.09	171%

## Data Availability

The genomic sequences obtained in this study, were included in the BioProject PRJNA719273 available at NCBI. The BioProject contains 70 BioSamples with IDs from SAMN18593889 to SAMN18593958.

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
