# Peer review of "Climatic Zone and Soil Properties Determine the Biodiversity of the Soil Bacterial Communities Associated to Native Plants from Desert Areas of North-Central Algeria"

_microorganisms, 2021, doi:10.3390/microorganisms9071359_

Round 1
Reviewer 1 Report
The work entitled “Climatic zone and soil properties determine the biodiversity of the soil microbial communities associated to native plants from desert areas of North-Central Algeria” by Bona et al. aims at contributing to the characterization of the microbial communities associated with several several soil types and autochthonous plants from Algeria.
The paper is generally well written though conclusions are at least debatable; data processing is good. The experimental design, and the number of samples collected are borderline to support some of the conclusions.
More information regarding sampling is needed to determine if the conclusions of this study are right or not. It is not clear if the soil that was analyzed was indeed associated with the root system. In arid areas the root system might go easily up to 50 cm in depth. Say nothing about Retama raetam which, as the authors mentioned, is used in soil fixation/dune stabilization! The authors mentioned that they dug pits to collect soil samples. How was the soil processed afterwards, taking into account that the root system of the 14 species analyzed was very different and only 0.25 g of soil was used for DNA isolation? Please provide relevant info such as the depth at which sampling was done, not only the fact that 5 cm from the surface was removed. What was the depth of the pits? The same for all species? What was the volume of the soil collected? Was the soil of each sample mixed?
Taking into consideration these limitations, the low number of biological samples analyzed (5), and the fact that there is very large variation of the alpha-diversity for most species analyzed (Figure 2), it is clear that this study characterizes more various types of soils rather than the microbial communities associated with native plants. However, this is not what readers can understand from the title. Changes throughout the MS have to be made to reflect the fact that sampling was quite limited to assess the influence of the plant root system on the soil microbial diversity.
Other comments:
Throughout the text references have to be reformatted. For example, instead of [5][6][7][8][9][10][11][12][13][14][15] the reference should be [5-15].
Line 48.
238 million km2 should be either ha or 2.38 milion Km2.
Line 120
“DNA of each sample is quantified by”. Use past tense – was.
Lines 123-126
The sentence could be written better. It is very unusual to say “you”: “This kit has two distinct PCRs: the first is a PCR target that allows you to select the V3-V4-V5 regions, the second a PCR index, which allows you to tie the bar-codes and adapters compatible with the Illumina MiSeq platform, used for pyrosequencing.”
Line 129 Table 1. This table summarizes interesting info, but this paper is not a review. If new data generated in this study is presented in the supplementary material then Table 1 should be moved there as well.
Lines 131-132
Please revise the sentence and make sure you deliver the right information “Bioinformatic analysis workflow was proposed in this paper by the authors for the first time.” Time usage is incorrect.
Lines 379-380
Please revise the sentence: “The present work deeply describes the microbial communities associated to the soils of different desert regions of North-Central Algeria.” This can be considered as a preliminary study, not an in-depth study of the microbial communities in the soils in such a large area.
Lines 459-465
The conclusions need revision. The 2 sentences cover topics that are too general.
Author Response
Responses to Reviewer 1
The work entitled “Climatic zone and soil properties determine the biodiversity of the soil microbial communities associated to native plants from desert areas of North-Central Algeria” by Bona et al. aims at contributing to the characterization of the microbial communities associated with several soil types and autochthonous plants from Algeria.
The paper is generally well written though conclusions are at least debatable; data processing is good. The experimental design, and the number of samples collected are borderline to support some of the conclusions.
More information regarding sampling is needed to determine if the conclusions of this study are right or not. It is not clear if the soil that was analyzed was indeed associated with the root system. In arid areas the root system might go easily up to 50 cm in depth. Say nothing about Retama raetam which, as the authors mentioned, is used in soil fixation/dune stabilization! The authors mentioned that they dug pits to collect soil samples. How was the soil processed afterwards, taking into account that the root system of the 14 species analyzed was very different and only 0.25 g of soil was used for DNA isolation? Please provide relevant info such as the depth at which sampling was done, not only the fact that 5 cm from the surface was removed. What was the depth of the pits? The same for all species? What was the volume of the soil collected? Was the soil of each sample mixed?
Before sampling, 5 cm topsoil was removed. Soils associated to the roots (rhizosphere soils) were collected perpendicularly all along the roots using sterile gloves to a depth ranging from 5 to 20 centimeter. The depth of pits was 15 cm of after topsoil removal and it was the same for all the species. The five pits (around 20x20x15 cm) were dug for each sampled plant, then the soil was mixed to form five biological replicas and stored at 4°C till laboratory, where the soils were maintained at -80°C until analyses. The final volume of each soil was around 100 cm3.
Retama raetam is among the plants widely used for dune stabilization and soil fixation in the Algerian steppe region which suffers from desertification and sand movements.
0.25 g was the amount of soil used for DNA extraction following the manufacturer’s instructions; the soil used for this analysis was sampled after carefully mixing the original sample.
Taking into consideration these limitations, the low number of biological samples analyzed (5), and the fact that there is very large variation of the alpha-diversity for most species analyzed (Figure 2), it is clear that this study characterizes more various types of soils rather than the microbial communities associated with native plants. However, this is not what readers can understand from the title.
As underlined also in the previous answer, the authors sampled rhizosphere soil and not simple soil, so what reported in the title is correct. In fact, as it is possible to observe in figure 3, the beta diversity of the communities associated to the different plant species, are quite different. However, using also other statistical approaches (e.g. PCA), the authors interpreted the factors sampling site and climatic zone as more relevant in determining the microbial community composition in respect to plant species.
Changes throughout the MS have to be made to reflect the fact that sampling was quite limited to assess the influence of the plant root system on the soil microbial diversity.
The demonstration of the fact that we sampled the rhizosphere soil and that the influence of the root system determined the changes in the different communities is evident from figure 3A where the communities associated with each plant are practically independent each other.
Other comments:
Throughout the text references have to be reformatted. For example, instead of [5][6][7][8][9][10][11][12][13][14][15] the reference should be [5-15].
Done.
Line 48.
238 million km2 should be either ha or 2.38 milion Km2.
Done.
Line 120
“DNA of each sample is quantified by”. Use past tense – was.
Done.
Lines 123-126
The sentence could be written better. It is very unusual to say “you”: “This kit has two distinct PCRs: the first is a PCR target that allows you to select the V3-V4-V5 regions, the second a PCR index, which allows you to tie the bar-codes and adapters compatible with the Illumina MiSeq platform, used for pyrosequencing.”
The authors deleted this sentence.
Line 129 Table 1. This table summarizes interesting info, but this paper is not a review. If new data generated in this study is presented in the supplementary material then Table 1 should be moved there as well.
The authors move table 1 to supplemental materials.
Lines 131-132
Please revise the sentence and make sure you deliver the right information “Bioinformatic analysis workflow was proposed in this paper by the authors for the first time.” Time usage is incorrect.
The authors modified the time usage.
Lines 379-380
Please revise the sentence: “The present work deeply describes the microbial communities associated to the soils of different desert regions of North-Central Algeria.” This can be considered as a preliminary study, not an in-depth study of the microbial communities in the soils in such a large area.
The authors revised the sentence.
Lines 459-465
The conclusions need revision. The 2 sentences cover topics that are too general.
The authors revised the conclusions.
Reviewer 2 Report
The paper tried to compare the microbial community of plant associated microbiome of 14 native plants in Algeria in two distinct climatic zone and three different soil conditions (mainly pH).
It is not clear from the introduction the criteria of the selection of the 14 plant species and also the sampling sites.
The other problem was how to differentiate the rhizosphere and non-rhizosphere soil and the plant was active or inactive during sampling?
I am not sure that the taxonomic resolution is really at species level, please clarify it.
Special comments:
In Table S1:
Please change CaCo3 to CaCO3. In last column plant species, instead of rhizosphere soil.
In Figure S1: What is the source of the meteorological data? Please cite it.
Author Response
Responses to Reviewer 2
The paper tried to compare the microbial community of plant associated microbiome of 14 native plants in Algeria in two distinct climatic zone and three different soil conditions (mainly pH).
It is not clear from the introduction the criteria of the selection of the 14 plant species and also the sampling sites.
These plants are the most widespread spontaneous plants in the sampling sites, that were their natural areals, this may refers to beneficial actors belonging microbiomes that help plants to survive in harsh environments characterized by abiotic stress conditions.
The other problem was how to differentiate the rhizosphere and non-rhizosphere soil and the plant was active or inactive during sampling?
After digging pits around the plants, the rhizosphere soil was differentiated by removing the external soil then soil found adjacent to the roots was collected using sterile gloves. The plants were active during sampling.
I am not sure that the taxonomic resolution is really at species level, please clarify it.
The sentence “at species level” indicates the attribution level at which the bioinformatic analysis was done. Unfortunately, the most organisms were attributed only at genus level, but in some cases the species level was reached. So if the authors have chosen the genus level, they would lost part of the information.
Special comments:
In Table S1:
Please change CaCo3 to CaCO3. In last column plant species, instead of rhizosphere soil.
Done.
In Figure S1: What is the source of the meteorological data? Please cite it.
The meteorological data related to the monthly average of maximum and minimum temperatures were identified by the two stations of Djelfa and Ghardaia of the Algerian National Office of Meteorology. ONM collect the climatic data by using automatic stations near to the sampling sites, the collected data were verified and compared to Alessandria’s climatic data using the Google Earth Engine tool.
The authors added this information in the text according to the reviewer suggestion.
Round 2
Reviewer 1 Report
The revised work entitled “Climatic zone and soil properties determine the biodiversity of the soil microbial communities associated to native plants from desert areas of North-Central Algeria” by Bona et al. addressed most of the suggestions in a constructive way.
There are a few things that still have to be addressed.
Major
- The title needs to accommodate some changes to reflect correctly to work described in the paper. The authors described correctly that plant microbiota comprises “different microbial species including archaea, bacteria, fungi, and protists (line 67)”. In order to characterize the diversity of these the soil microbial communities on have to sequence 16S (prokaryotes), 18S (eukaryotes) and ITS (fungi) amplicons. However, the authors sequenced only 16S amplicons so they did not characterize the soil microbial communities but soil bacterial communities. As a result, the title of the paper is incorrect. The authors should change “soil microbial communities” to “soil bacterial communities”.
- The conclusions are still not up to what it should be. For example, the sentence “Despite the plant roots selected their own microbial populations, this work showed that in stressed environments, such as desert, the climatic conditions and the composition of the substrate strongly influenced the selection of bacterial populations (lines 463-465)” should be reformulated as i). it is confusing (was the influence of the substrate and of climatic conditions stronger than that of species/roots?); ii) time correspondence in this sentence is not correct. Also, please reformulate the last sentence and find an alternative word for “precise” in “providing precise information still missing” (line 467).
Minor
Line 48. Km should be km (no capital letter)
Lines 63-65. Please reformulate the sentence. Why “in a country”?
Line 85. Replace “microbiota” with “bacterial communities”.
Lines 103-104. The sentence has to be revised: “and stored at 4°C till laboratory, where the soils were maintained at -80°C until analyses.” Consider something like “stored at 4°C until taken to laboratory” and “stored” instead of “maintained”.
Author Response
Reviewer 1
The revised work entitled “Climatic zone and soil properties determine the biodiversity of the soil microbial communities associated to native plants from desert areas of North-Central Algeria” by Bona et al. addressed most of the suggestions in a constructive way.
There are a few things that still have to be addressed.
Major
- The title needs to accommodate some changes to reflect correctly to work described in the paper. The authors described correctly that plant microbiota comprises “different microbial species including archaea, bacteria, fungi, and protists (line 67)”. In order to characterize the diversity of these the soil microbial communities on have to sequence 16S (prokaryotes), 18S (eukaryotes) and ITS (fungi) amplicons. However, the authors sequenced only 16S amplicons so they did not characterize the soil microbial communities but soil bacterial communities. As a result, the title of the paper is incorrect. The authors should change “soil microbial communities” to “soil bacterial communities”.
The authors changed the title according to the suggestion.
- The conclusions are still not up to what it should be. For example, the sentence “Despite the plant roots selected their own microbial populations, this work showed that in stressed environments, such as desert, the climatic conditions and the composition of the substrate strongly influenced the selection of bacterial populations (lines 463-465)” should be reformulated as i). it is confusing (was the influence of the substrate and of climatic conditions stronger than that of species/roots?); ii) time correspondence in this sentence is not correct. Also, please reformulate the last sentence and find an alternative word for “precise” in “providing precise information still missing” (line 467).
The authors changed the conclusions taking into account the reviewer suggestion.
Minor
Line 48. Km should be km (no capital letter). Done.
Lines 63-65. Please reformulate the sentence. Why “in a country”? Done.
Line 85. Replace “microbiota” with “bacterial communities”. Done.
Lines 103-104. The sentence has to be revised: “and stored at 4°C till laboratory, where the soils were maintained at -80°C until analyses.” Consider something like “stored at 4°C until taken to laboratory” and “stored” instead of “maintained”. Done.